# Adam on Local Time: Addressing Nonstationarity in RL with Relative Adam Timesteps

**Benjamin Ellis**[*]
University of Oxford

**Matthew T. Jackson**[*]
University of Oxford

**Andrei Lupu**
University of Oxford

**Alexander D. Goldie**
University of Oxford

**Mattie Fellows**
University of Oxford

**Shimon Whiteson**
University of Oxford

**Jakob N. Foerster**
University of Oxford

## Abstract

In reinforcement learning (RL), it is common to apply techniques used broadly in machine learning such as neural network function approximators and momentum-based optimizers [1, 2]. However, such tools were largely developed for supervised learning rather than nonstationary RL, leading practitioners to adopt target networks [3], clipped policy updates [4], and other RL-specific implementation tricks [5, 6] to combat this mismatch, rather than directly adapting this toolchain for use in RL. In this paper, we take a different approach and instead address the effect of nonstationarity by adapting the widely used Adam optimiser [7]. We first analyse the impact of nonstationary gradient magnitude—such as that caused by a change in target network—on Adam's update size, demonstrating that such a change can lead to large updates and hence sub-optimal performance. To address this, we introduce *Adam with Relative Timesteps*, or Adam-Rel. Rather than using the global timestep in the Adam update, Adam-Rel uses the local timestep within an epoch, essentially resetting Adam's timestep to 0 after target changes. We demonstrate that this avoids large updates and reduces to learning rate annealing in the absence of such increases in gradient magnitude. Evaluating Adam-Rel in both on-policy and off-policy RL, we demonstrate improved performance in both Atari and Craftax. We then show that increases in gradient norm occur in RL in practice, and examine the differences between our theoretical model and the observed data.

## 1 Introduction

Reinforcement Learning (RL) aims to learn robust policies from an agent's experience. This has the potential for large scale real-world impact in areas such as autonomous driving or improving logistic chains. Over the last decade, a number of breakthroughs in supervised learning—such as convolutional neural networks and the Adam optimizer—have expanded the deep learning toolchain and been transferred to RL, enabling it to begin fulfilling this potential.

However, since RL agents are continuously learning from new data they collect under their changing policy, the optimisation objective is fundamentally *nonstationary*. Furthermore, temporal difference (TD) approaches bootstrap the agent's update from its own value predictions, exacerbating the nonstationarity in the objective function. This is in stark contrast to the *stationary* supervised learning setting for which the deep learning toolchain was originally developed. Therefore, to apply these tools successfully, researchers have developed a variety of implementation tricks *on top of* this base to stabilise training [8, 6, 5]. This has resulted in a proliferation of little-documented design choices that are vital for performance, contributing to the reproducibility crisis in RL [9].

---

[*]Equal Contribution

38th Conference on Neural Information Processing Systems (NeurIPS 2024).

We believe that in the long term, a more robust approach is to *augment* this toolchain for RL, rather than building on top of it. To this end, in this paper we examine the interaction between nonstationarity and the Adam optimizer [7]. Adam's update rule, where equations are applied element-wise (i.e. per parameter), is defined by

$$m_t = \beta_1 m_{t-1} + (1 - \beta_1)g_t, \qquad \hat{m}_t = \frac{m_t}{(1 - {\beta_1}^t)},$$

$$v_t = \beta_2 v_{t-1} + (1 - \beta_2){g_t}^2, \qquad \hat{v}_t = \frac{v_t}{(1 - {\beta_2}^t)},$$

$$u_t = \frac{\hat{m}_t}{\sqrt{\hat{v}_t} + \epsilon}, \qquad \theta_t = \theta_{t-1} - \alpha u_t.$$

Here, $g_t$ is the gradient, $\theta_t$ a parameter to be optimized, and $\alpha$ the learning rate. The resulting update is the ratio of two different momentum terms: one for the first moment, $m_t$, and one for second moment, $v_t$, of the gradient. These terms use different exponential decay coefficients, $\beta_1$ and $\beta_2$. Under stationary gradients, the $(1 - \beta_i)$ weighting ensures that, in the limit, the overall magnitude of the two momenta is independent of the value chosen for each of the coefficients. However, since both momentum estimates are initialised to 0, they must be renormalised for a given (finite) timestep $t$, to account for the "missing parts" of the geometric series [7], resulting in $\hat{v}_t$ and $\hat{m}_t$.

Crucially, $t$ counts the update steps since the *beginning of training* and thus bakes in the assumption of stationarity that is common in supervised learning. In particular, this renormalisation breaks down if the loss is nonstationary. Consider a task change late in training, which results in gradients orders of magnitudes higher than those of the prior (near convergence) task. Clearly, this is analogous to the situation at the *beginning of training* where all momentum estimates are 0. However, the $t$ parameter, and therefore the renormalisation, does not account for this.

In this paper, we demonstrate that changes in the gradient scale can lead to large updates that persist over a long horizon. Previous work [10, 11] has suggested that old momentum estimates can *contaminate* an agent's update and propose resetting the entire optimizer state when the target changes as a solution. However, by discarding previous momentum estimates, we hypothesise that this approach needlessly sacrifices valuable information for optimization. Instead, we propose retaining momentum estimates and only resetting $t$, which we refer to as **Adam-Rel**. In the limit of gradient sparseness, we show that the Adam-Rel update size remains bounded, converging to 1 in the limit of a large gradient, unlike Adam. Furthermore, if such gradient magnitude increases do not occur, Adam-Rel reduces to learning rate annealing, a common method for stabilising optimization.

When evaluated against the original Adam and Adam with total resets, we demonstrate that our method improves PPO's performance in Craftax-Classic [12] and the Atari-57 challenge from the Arcade Learning Environment [13]. Additionally, we demonstrate improved performance in the off-policy setting by evaluating DQN on the Atari-10 suite of tasks [14]. We then examine the gradients in practice and show that there are significant increases in gradient magnitude following changes in the objective. Finally, we examine the discrepancies between our theoretical model and observed gradients to better understand the effectiveness of Adam-Rel.

## 2 Background

### 2.1 Reinforcement Learning

**Definition** Reinforcement learning agents learn a policy $\pi$ in a Markov Decision Process [15, MDP], a tuple $M = \langle \mathcal{S}, \mathcal{A}, \mathcal{T}, \mathcal{R}, \gamma \rangle$ where $\mathcal{S}$ is the set of states, $\mathcal{A}$ is the set of actions, $\mathcal{T} : \mathcal{S} \times \mathcal{A} \to \mathcal{P}(\mathcal{S})$ is the transition function, $\mathcal{R} : \mathcal{S} \times \mathcal{A} \to \mathbb{R}$ is the reward function and $\gamma$ is the discount factor. At each timestep $t$, the agent observes a state $s_t \in \mathcal{S}$ and takes an action $a_t$ drawn from $\pi(\cdot|s_t)$ before transitioning to a new state $s_{t+1} \in \mathcal{S}$ and receiving reward $r_t$ drawn from $\mathcal{R}(s_t, a_t)$. The goal of the agent is to maximise the expected discounted return $\mathbb{E}_{\pi, \mathcal{T}} \left[ \sum_{t=0}^{\infty} \gamma^t r_t \right]$.

**Nonstationarity in RL** In contrast with supervised learning, where a single stationary objective is typically optimised, reinforcement learning is inherently nonstationary. Updates to the policy induce changes not only in the distribution of observations seen at a given timestep, but also the return distribution, and hence value function being optimised. This arises regardless of how these updates

are performed. However, one particular reason for nonstationarity in RL is the use of bootstrapped value estimates via TD learning [15], which optimises the below objective

$$\mathcal{L}(\theta) = \left[ \text{sg} \left\{ r_t + \gamma V_\theta^\pi \left( s_{t+1} \right) \right\} - V_\theta^\pi \left( s_t \right) \right]^2,$$

where sg is the stop-gradient operator. In this update, the target $r_t + \gamma V_\theta^\pi(s_{t+1})$ depends on the parameters $\theta$ and therefore changes as these are updated.

These target changes can either be more gradual, as in the case of continuous updates to the value function in TD learning, or more abrupt, as in the case of the use of target networks in DQN.

**Sequentially Optimized Stationary Objectives**   In this work, we focus on abrupt objective changes; changes of objectives that do not involve a smoothing method such as Polyak averaging [1], and the resulting sudden change of supervised learning problem. More explicitly, we consider optimising a stationary loss function $L(\theta, \phi)$, where $\theta$ are the parameters to be optimised and $\phi$ is the other parameters of the loss function (such as the parameters of a value network), which are not updated throughout optimisation, but does not include the training data.

We consider a setting where at a certain timestep $t$ in our training, we transition from optimising $L(\theta_t, \phi_1)$ to optimising $L(\theta_{t+1}, \phi_2)$ for some $\phi_1$, $\phi_2$. Such individual objectives are still non-stationary. For example, significant changes in the policy would induce changes in the data distribution, which would then affect the underlying loss landscape, but we do not consider such non-stationarity in this work.

This setting is very common throughout RL. Bootstrapped value estimates are the most common cause of this, but it also occurs in PPO's actor update, where each new rollout induces a different supervised learning problem due to the actor and critic updates. This is optimised for a fixed number of updates before collecting new data.

We refer to these sequences of supervised learning problems as sequentially-optimised stationary objectives. In this work, we use this framing to propose an approach that is consistent throughout each stationary period of optimization and applies corrections to make optimization techniques valid when nonstationarity is introduced via objective changes. Bengio et al. [11] propose the gradient contamination hypothesis, which states that current optimizer momentum estimates can point in the opposite direction to the gradient following a change in objective, thereby hindering optimization. A previous approach to this problem is that of Asadi et al. [10], where they propose resetting Adam's momentum estimates and timestep to $0$ throughout training. We refer to this method as **Adam-MR**. Finally, Dohare et al. [16] propose setting the Adam hyperparameters to equal values, such that $\beta_1 = \beta_2$, suggesting that this can help avoid performance collapse.

**Proximal Policy Optimization**   Proximal Policy Optimization [4, PPO] is a policy optimisation based RL method. It uses a learned critic $V_\phi^\pi$ trained by a TD loss to estimate the value function, and a clipped actor update of the form

$$\min \left[ \text{clip} \left( r_{(\theta,t)}, 1 \pm \epsilon \right) A^\pi(s_t, a_t), r_{(\theta,t)} A^\pi(s_t, a_t) \right], \tag{1}$$

where the policy ratio $r_{(\theta,t)} = \frac{\tilde{\pi}_\theta(a_t|s_t)}{\pi(a_t|s_t)}$ is the ratio of the stochastic policy to optimise $\tilde{\pi}_\theta$ and $\pi$, the previous policy. $A^\pi$ is the advantage, which is typically estimated using generalised advantage estimation [17]. Clipping the policy ratio aims to avoid performance collapse by preventing policy updates larger than $\epsilon$.

Optimisation of the PPO objective proceeds by first rolling out the policy to collect data, and then iterating over this data in a sequence of *epochs*. Each of these epochs splits the collected data into a sequence of *mini-batches*, over which the above update is calculated.

## 2.2   Momentum-Based Optimization

Momentum [1, 2] is a method for enhancing stochastic gradient descent by accumulating gradients in the direction of repeated improvement. The typical formulation of momentum for each element $i$ is

$$m_t^i = \beta m_{t-1}^i + g_t^i,$$
$$\theta_t^i = \theta_{t-1}^i - \alpha m_t^i,$$

where $\beta$ is the momentum coefficient, $g_t \in \mathbb{R}^n$ is the gradient at the current step, $m_t \in \mathbb{R}^n$ is the gradient incorporating momentum, $\alpha$ is the scalar learning rate and $\theta \in \mathbb{R}^n$ are the parameters to be optimised. With momentum, update directions with low curvature have their contribution to the gradient amplified, considerably reducing the number of steps required for convergence.

In the introduction, we described the update equations for Adam [7], the most popular optimizer that uses momentum. Adam's update is designed to keep its updates within a trust region, which depends on a learning rate $\alpha$.

## 3 Nonstationary Optimization with Adam

We now investigate the effect of nonstationarity on Adam by analysing its update rule after a sudden change in gradient. As a simplified model of gradient instability, we assume optimization with Adam starts at timestep $t = -t'$ with a constant gradient $g^i_{-t'} = g$, $0 < g < \infty$ until timestep 0. Following $t = 0$, we model instability by increasing the gradient by a factor of $k$, as might occur in a nonstationary optimization setting. This gives

$$g^j_t = \begin{cases} g, & -t' \leq t < 0, \\ kg, & t \geq 0. \end{cases} \tag{2}$$

For larger values of $t'$, the short term effects of Adam's initialisation on the momentum terms dissipate and $\hat{m}_t$ and $\hat{v}_t$ converge to stable values. By taking the limit of $t' \to \infty$, we investigate the effect of a sudden change in gradient $g^i_t$ on the update size $u^i_t$ after a long period of training. This allows for any effects from the initialisation of momentum terms $\hat{m}_{-t',t}$ and $\hat{v}_{-t',t}$ to dissipate:

**Theorem 3.1.** *Assume that $\epsilon = 0$. Let $g^i_t$ be defined as in Equation* (2) *and $\hat{m}^i_{-t',t}$ and $\hat{v}^i_{-t',t}$ be the momentum terms at timestep $t$ given Adam starts at timestep $-t'$. It follows that:*

$$\lim_{t' \to \infty} u^i_t = \lim_{t' \to \infty} \frac{\hat{m}^i_{-t',t}}{\sqrt{\hat{v}^i_{-t',t}}} = \frac{\beta_1^{t+1} + k(1 - \beta_1^{t+1})}{\sqrt{\beta_2^{t+1} + k^2(1 - \beta_2^{t+1})}}. \tag{3}$$

*Proof.* See Appendix A. $\qquad\square$

For large $k$, Theorem 3.1 proves that the element-wise momentum term after the change in gradient at $t = 0$ is approximately $\frac{1 - \beta_1}{\sqrt{1 - \beta_2}}$. For the most commonly used values of $\beta_1 = 0.9$ and $\beta_2 = 0.999$, this is $\sqrt{10}$, which is much larger than the intended unit update which Adam is designed to maintain. The top plot in Figure 1, which shows the Adam update size against $t$ for different values of $k$, demonstrates that the update peaks significantly higher than the desired 1 before slowly converging back to 1.

## 4 Adam with Relative Timesteps

To fix the problems analysed in the previous section, we introduce Adam-Rel. At the start of each new supervised learning problem, Adam-Rel resets Adam's $t$ parameter to 0, rather than incrementing it from its previous value. This one-line change is illustrated for PPO in Algorithm 1.

At the start of training, both momentum terms in Adam are 0. Therefore, at the first timestep, when the first gradient is encountered, the magnitude of the gradient is infinite relative to the current momentum estimate. As explained in Section 3, this induces a large update. However, dividing the momentum estimates by $(1 - \beta_1^t)$ and $(1 - \beta_2^t)$ fixes this issue by correcting for this sparsity. Therefore, by resetting $t$ to 0, Adam handles changes in gradient magnitude resulting from the change of supervised learning problem.

If we examine the same update as in the previous section adjusted by Adam-Rel, assuming that we reset Adam's $t$ just before the gradient scales to $kg$, we find it comes to

$$\lim_{t' \to \infty} \frac{\hat{m}^i_{-t',t}}{\sqrt{\hat{v}^i_{-t',t}}} = \frac{\sqrt{1 - \beta_2^{t+1}}}{1 - \beta_1^{t+1}} \frac{\beta_1^{t+1} + k(1 - \beta_1^{t+1})}{\sqrt{\beta_2^{t+1} + k^2(1 - \beta_2^{t+1})}}. \tag{4}$$

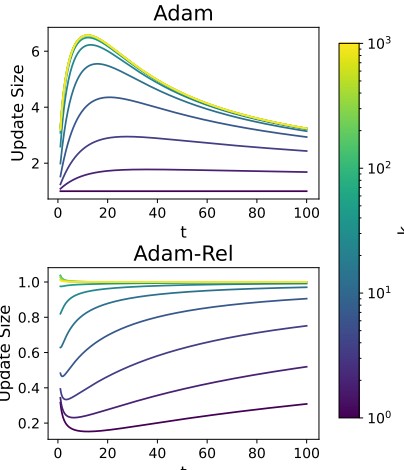

Figure 1: Update size of Adam and Adam-Rel versus $k$ when considering nonstationary gradients. Assumes that optimization starts at time $-t'$, which is large, and that the gradients up until time 0 are $g$ and then there is an increase in the gradient to $kg$.

**Algorithm 1** Pseudocode for PPO with Adam, Adam-Rel, and Adam-MR.

$m_0 = 0, v_0 = 0, t = 0$       ▷ Initialise Adam state
$j = 0$       ▷ Initialise number of updates to 0
**for** $k = 1$ **to** $K$ **do**
    Rollout policy $\pi_{\theta_k}$ to collect batch $D$
    $t = 0$
    $t = 0, m_j = 0, v_j = 0$
    **for** epoch $= 1$ **to** $E$ **do**
        **for** mini-batch $B$ in $D$ **do**
            $g_j = \nabla_{\theta_{j-1}}\left[L^{\text{PPO}}(\theta_{j-1}) + L^{\text{TD}}(\theta_{j-1})\right]$
            $j = j + 1$
            $t = t + 1$
            $m_j = \beta_1 m_{j-1} + (1 - \beta_1)g_j$
            $v_j = \beta_2 v_{j-1} + (1 - \beta_2)g_j{}^2$
            $\hat{m}_j = \frac{m_j}{(1 - \beta_1{}^t)}$
            $\hat{v}_j = \frac{v_j}{(1 - \beta_2{}^t)}$
            $\theta_j = \theta_{j-1} - \alpha\frac{\hat{m}_j}{\sqrt{\hat{v}_j} + \epsilon}$
        **end for**
    **end for**
**end for**

As $k \to \infty$, this tends to 1. This means that Adam-Rel ensures approximately unit update size in the case of a large increase in magnitude in the gradient, at the expense of a potentially smaller update at the point $t$ is reset. Figure 1 shows the update size of Adam-Rel as $t - t'$ increases. The update size is smaller at the start, but never reaches significantly above 1.

However, the above analysis does not show how Adam and Adam-Rel differ in practice, where large changes in gradient magnitude may not occur. Examining the bottom of Figure 1, we can see that for lower values of $k$, Adam-Rel rapidly decays the update size before increasing it. Functionally, this behaves like a learning rate schedule. Over a short horizon (e.g., 16 steps is common in PPO), this effect is similar to learning rate annealing, whilst over a longer horizon (e.g., approximately 1000 steps in DQN) it is akin to learning rate warmup, both of which are popular techniques in optimising stationary objectives. Therefore, the benefits of Adam-Rel are twofold: first, it guards against large increases in gradient magnitude by capping the size of potential updates, and secondly, if such large gradient increases do not occur, it reduces to a form of learning rate annealing, which is commonly employed in optimising stationary objectives.

## 5 Experiments

### 5.1 Experimental setup

To evaluate Adam-Rel, we explore its impact on DQN and PPO, two of the most popular algorithms in off-policy and on-policy RL respectively.

To do so, we first train DQN [18, 19] agents with Adam-Rel on the Atari-10 benchmark for 40M frames, evaluating performance against agents trained with Adam and Adam-MR. We then extensively evaluate our method's impact on PPO [4, 19, 20], training agents on Craftax-Classic-1B [12]—a JAX-based reimplementation of Crafter [21] where the agent is allocated 1 billion environment interactions—and the Atari-57[2] suite [13] for 40 million frames. In doing so, our benchmarks respectively evaluate the performance of Adam-Rel on exceedingly long training horizons and its

---

[2]We exclude 2 out of the 57 games, Montezuma's Revenge and Venture, after observing that all algorithms achieve a human-normalized score of 0.

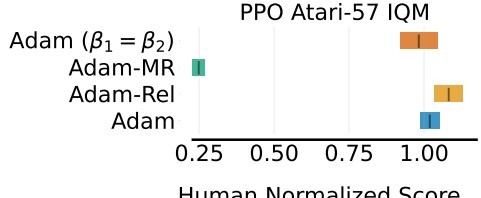 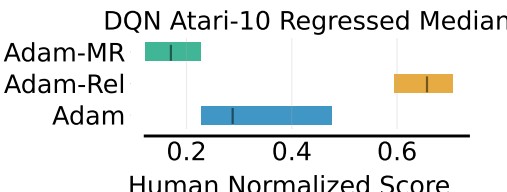

Figure 2: Performance of Adam-Rel, Adam, Adam-MR, and Adam ($\beta_1 = \beta_2$) for PPO and Adam, Adam-MR and Adam-Rel for DQN on Atari-57 and Atari-10 respectively. Atari-10 uses a subset of Atari tasks to estimate median performance across the whole suite. Details can be found in [14]. Error bars are 95% stratified bootstrapped confidence intervals. Results are across 10 seeds except for Adam ($\beta_1 = \beta_2$), which is 3 seeds.

robustness when applied to a diverse range of environments. We then analyse the differences between Adam-Rel and Adam's updates. We compare 8 seeds on the Craftax-Classic environment for this purpose, recording the update norm, maximum update, and gradient norm of every update.

## 5.2 Off-policy RL

Figure 2 shows the performance of DQN agents trained with Adam-Rel against those trained with Adam-MR and Adam on the Atari-10 benchmark [14]. We tune the learning rate of each method, keeping all other hyperparameters fixed at values tuned for Adam in CleanRL [19]. Adam-Rel outperforms Adam, achieving 65.7% vs. 28.8% human-normalized performance. Furthermore, the stark performance difference between Adam-Rel and Adam-MR (23.5%) demonstrates the advantage of retaining momentum information across target changes (so long as appropriate corrections are applied), thereby contradicting the gradient contamination hypothesis discussed in Bengio et al. [11] and Asadi et al. [10].

More surprisingly, Adam-MR performs substantially worse than Adam, contrasting with the findings of Asadi et al. [10]. We evaluate on a different set of Atari games and tune both Adam and Adam-MR separately, which may account for the differences. However, these results suggest that preventing any gradient information from crossing over target changes is an excessive correction and can even harm performance. We additionally evaluate on the set of games used by Asadi et al. [10], the results of which can be found in Appendix B. We find that Adam-Rel outperforms the Adam baseline in IQM. We also find that, although our implementation of Adam-MR again significantly under-performs relative to the Adam baseline, we approximately match the returns listed in their work.

We also evaluate Adam-Rel when soft target changes are used, by comparing Adam and Adam-Rel on Atari-10 when using DQN with Polyak averaging. We find that Adam-Rel also outperforms Adam in this setting. These results, along with a more detailed discussion, can be found in Appendix C.

## 5.3 On-policy RL

**Craftax**  Figure 3 shows the performance of PPO agents trained on Craftax-1B over 8 seeds. Most strikingly, Adam-MR, which resets the optimizer completely when PPO samples a new batch, achieves dramatically poorer performance across all metrics. This deficit is unsurprising when compared to its performance on DQN, where the optimizer has many more updates between resets and so can achieve a superior momentum estimate, and demonstrates the impact of not retaining any momentum information after resets in on-policy RL. Similarly, Adam with $\beta_1 = \beta_2$ [16] achieves poorer performance than Adam-Rel on all metrics and has no significant different against Adam with default hyperparameters.

Furthermore, Adam-Rel outperforms Adam on all metrics. Whilst the performance on the number of achievements is similar, we follow the evaluation procedure recommended in Hafner [21] and report score, calculated as the geometric mean of success rates for all achievements. This metric applies logarithmic scaling to the success rate of each achievement, thereby giving additional weight to those that are hardest to accomplish. We see that Adam-Rel clearly outperforms Adam in score, as well as on the two hardest achievements (collecting diamonds and eating a plant). These behaviours require

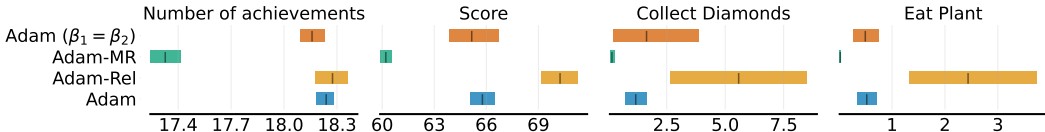

Figure 3: PPO on Craftax-1B — comparison of Adam-Rel against Adam, Adam-MR, and Adam with $\beta_1 = \beta_2$ [16]. Bars show the 95% stratified bootstrap confidence interval, with mean marked, over 8 seeds [22].

a strong policy to discover so are learned late in training, suggesting that Adam-Rel improves the plasticity of PPO.

**Atari-57**   Figure 2 shows the performance of PPO agents on Atari-57. As before, entirely resetting the optimizer significantly harms performance when compared to resetting only the count. Across all environments, Adam-Rel also improves over Adam, outperforming it in **33 out of the 55 games** tested and IQM across games. Adam with $\beta_1 = \beta_2$ also fails to improve over the baseline.

To further analyse the impact of Adam-Rel over Atari-57, we plot the performance profile of human-normalized score (Figure 4). Whilst the performance of the two methods is similar over the bottom half of the profile, we see a major increase in performance in the top half. Namely, at the 75th percentile of scores Adam-Rel achieves a human-normalized performance of **338% vs. 220%** achieved by Adam. This demonstrates the ability of Adam-Rel to improve policy per-

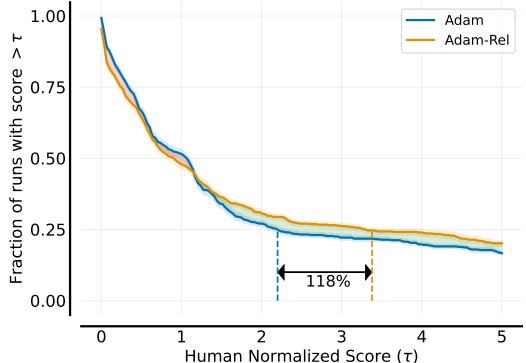

Figure 4: Performance Profile of Adam and Adam-Rel on Atari-57. Error bars represent the standard error across 10 seeds. Green-shaded areas represent Adam-Rel outperforming Adam and red-shaded areas the opposite.

formance on tasks where Adam is successful but suboptimal, without sacrificing performance on harder tasks.

## 5.4   Method Analysis

In this section we connect our theoretical exposition in Section 3 to our experimental results. Specifically, we first examine whether gradients increase in magnitude due to nonstationarity, to what extent predictions from our model match the resulting updates, and how Adam's update differs from Adam-Rel's in practice.

To this end, we collect gradient (i.e., before passing through the optimizer) and update (i.e., the final change applied to the network) information from PPO on Craftax-Classic. We follow the experimental setup in Section 5 but truncate the Craftax-Classic runs to 250M steps to reduce the data processing required. The results are shown in Figure 5.

**Comparing Theory and Practice**   In Figure 5, both Adam and Adam-Rel face a significant increase in gradient norm immediately after starting optimisation on a new objective resulting from a new batch of trajectories collected under an updated policy and value function. While this matches the assumptions we make in our work, the magnitude of the increase is much less than some of the values explored in Section 3.

For Adam, this is approximately 29% and for Adam-Rel it is around 45%. The grad norm profiles look similar in each case, with the norm peaking early before decreasing below its initial average value. This decrease and the initial ramp both deviate from the step function we assume in our model. It is obvious that our theoretical model of gradients, which requires an increase in the gradient magnitude on each abrupt change in the objective, cannot hold throughout training in its entirety because this would require the gradient norm to increase without bound.

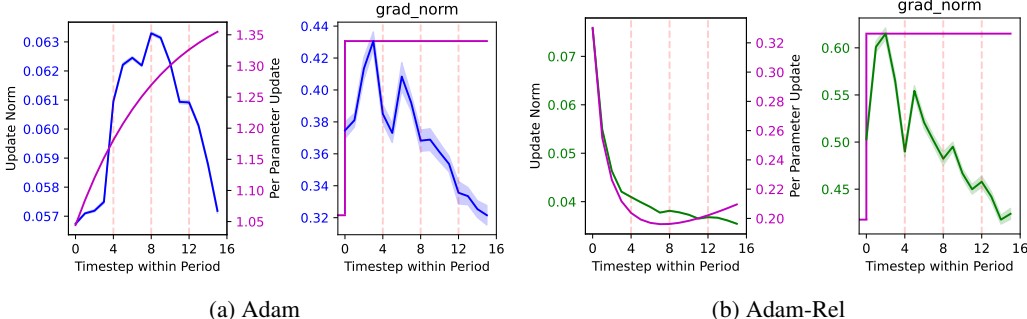

(a) Adam             (b) Adam-Rel

Figure 5: Adam and Adam-Rel compared to the theoretical model. To make this plot, we divided all the updates in the PPO run into chunks, each of which was optimising a stationary objective. We then averaged over all the chunks. The red dashed lines show the different epochs for each batch of data. The assumption about the gradient under the model is shown in the grad norm plot. Note that the update norm plot for Adam and Adam-Rel has separate y-axes. The shading represents standard error.

However, we find that despite this discrepancy, for Adam-Rel the update predicted by our model fairly closely matches the shape of the true update norm, i.e., a *fast drop* at the beginning followed by flattening (the scaling is not comparable between observed and predicted values).

For Adam, our model explains the initial *overshoot* of the update norm but then fails to predict the rapid decrease, which results from the fast drop in the true gradient norm. Given the simplicity of our modeling assumptions, we find these results overall encouraging.

**On Spherical Cows** Under the assumption of a *step increase* in gradients of an *infinite* relative magnitude Adam-Rel results in a flat update, while Adam would drastically overshoot. Clearly, this assumption does not hold in practice, as we have shown above. However, we believe that this mismatch between reality and assumption is encouraging, since our experimental results show that Adam-Rel is still effective in this regime. Our hypothesis is that there are two benefits to designing Adam-Rel under these assumptions. First of all, it avoids overshoots even under large gradient steps and secondly, when there are less drastic gradient steps it *undershoots*, which might have similar effects to a fast learning rate annealing. These kind of annealing schedules (over longer horizons) are popular when optimising stationary losses [23, 24].

## 6 Related Work

**Optimization in Reinforcement Learning** Plasticity loss [25–27] refers to the loss in ability of models to fit new objectives as they are trained. This is particularly relevant in nonstationary settings such as RL and continual learning, where the model is continuously fitting changing objectives. Many solutions have been proposed, including resetting network layers [28–32], policy distillation [25], LayerNorm [33, 34], regressing outputs to their initial values [26], resetting dead units [35] and adding output heads during training [36]. These solutions, in particular resetting layers during training [28, 32], have contributed towards state-of-the-art performance on Atari 100k [30]. However, of these works, only Lyle et al. [33] investigate the relationship between the optimizer and nonstationarity, demonstrating that by reducing the momentum coefficient of the second-moment gradient estimate in Adam, the fraction of dead units no longer increases. However, these works focus on plasticity loss, which is a symptom of nonstationarity, and only analyse off-policy RL. In contrast, we address nonstationarity directly and evaluate both on-policy and off-policy RL.

Meta-reinforcement learning [37–39] provides an alternative approach to designing optimizers for reinforcement learning. Rather than manually identifying problems and handcrafting solutions for RL optimization, this line of work seeks to automatically discover these solutions by meta-learning components of the optimization process. Often these methods parameterize the agent's loss function with a neural network, allowing it to be optimized through meta-gradients [40–42] or zeroth-order methods [43, 20, 44]. Recently, Lan et al. [45] proposed meta-learning a black-box optimizer directly,

demonstrating competitive performance with Adam on a range of RL tasks. However, these works are limited by the distribution of tasks they were trained on, and using handcrafted optimizers in RL is still far more popular.

**Adam Extensions**  Cyclical update schedules [46] have previously been applied in supervised learning as a mechanism for simplifying hyperparameter tuning and improving performance, and Loshchilov and Hutter [47] propose the use of warm learning rate restarts with cosine decay for improving the training of convolutional nets. Liu et al. [48] examine the combination of Adam and learning rate warmup, proposing RAdam to stabilise training. However, all of these methods focus on supervised learning and therefore assume stationarity.

There has also been some investigation of the interaction between deep RL and momentum-based optimization. Henderson et al. [49] investigate the effects of different optimizer settings and recommend sensible parameters, but do not investigate resetting the optimizer. Bengio et al. [11] identify the problem of contamination of momentum estimates and propose a solution based on a Taylor expansion. Dohare et al. [16] investigate policy collapse in RL when training for longer than methods were tuned for and propose setting $\beta_1 = \beta_2$ to address this. By contrast, we investigate training for a standard number of steps and focus on improved overall empirical performance, rather than avoiding policy collapse. Asadi et al. [10], which is perhaps the most similar to our work, also aim to tackle contamination, but do so differently, by simply resetting the Adam momentum states to 0 whenever the target network changes in the value-based methods DQN and Rainbow. However, they do not consider resetting of Adam's timestep parameter, and explain their improved results by suggesting that old, bad, momentum estimates contaminate the gradients when training on a new objective. By contrast, we demonstrate that resetting only the timestep suffices for better performance on a range of tasks and therefore that the contamination hypothesis does not explain the better performance of resetting the optimizer. We also demonstrate that retaining momentum estimates can be essential for performance, particularly in on-policy RL.

**Adam in RL**  To adapt Adam for use in RL, prior work has commonly applied a number of modifications compared to its use in supervised learning [8]. The first is to set the parameter $\epsilon$ to $10^{-5}$, which is a higher value than the $10^{-8}$ typically used in supervised learning. Additionally many reinforcement learning algorithms use gradient clipping before passing the gradients to Adam. Typically gradient vectors are clipped by their $L_2$ norm.

A higher value of $\epsilon$ reduces the sensitivity of the optimizer to sudden large gradients. If an objective has been effectively optimized and hence the gradients are very small, then a sudden target change may lead to large gradients. $\hat{v}$ typically updates much more slowly than $\hat{m}$ and therefore this causes the update size to increase significantly, potentially causing performance collapse. However, this implementation detail is not mentioned in the PPO paper [4], and subsequent investigations omit it [6, 5]. Clipping the gradient by the norm also aims at preventing performance collapse. Andrychowicz et al. [6] find this to increase performance slightly when set to $0.5$.

## 7  Limitations and Future Work

In this work we have mostly examined *abrupt* nonstationarity, where there are distinct changes of target, as it is in that setting where our method can be most cleanly applied. However, a range of RL methods face *continuous* nonstationarity, such as when applying Polyak averaging [1] to smoothly update target networks after every optimization step. We have demonstrated improved performance in this algorithm in Appendix C. However, further investigation into how to apply resetting in this setting would be beneficial.

There are also many promising avenues for future work. First, while we have focused on RL, it would be interesting to apply Adam-Rel to other domains that feature nonstationarity such as RLHF, training on synthetic data, or continual learning. Additionally, we also note that while our results are promising, it was not possible to investigate all RL settings and environments in this work, and we therefore encourage future work in settings such as continuous control. Secondly, Adam-Rel is designed with the principle that large updates can harm learning, but it is not clear in general what properties of update sizes are desirable in nonstationary settings. Understanding this more clearly may help produce meaningful improvements in optimisation. Relatedly, it would be beneficial to better understand the nature of gradients in RL tasks, in particular how they change throughout training for

different methods and what effect this has on performance. Finally, re-examining other aspects of the RL toolchain that are borrowed from supervised learning could produce further advancements by designing architectures, optimisers and methods specifically suited for problems in RL.

## 8  Conclusion

We presented a simple, theoretically-motivated method for handling nonstationarity via the Adam optimizer. By analysing the impact of large changes in gradient size, we demonstrated how directly applying Adam to nonstationary problems can lead to unstable update sizes, before demonstrating how timestep resetting corrects for this instability. Following this, we performed an extensive evaluation of Adam-Rel against Adam and Adam-MR in both on-policy and off-policy settings, demonstrating significant empirical gains. We then demonstrated that increases in gradient magnitude after abrupt objective changes occur in practice and compared the predictions of our simple theoretical model with the observed data in a complex environment. Adam-Rel can be implemented as a simple, single-line extension to any Adam-based algorithm with discrete nonstationarity (e.g. target network updates), leading to major improvements in performance across environments and algorithm classes. We hope that the ease of implementation and effectiveness of Adam-Rel will encourage researchers to use it as a de facto component of future RL algorithms, providing a step towards robust and performant RL.

## Acknowledgements

BE and MJ are supported by the EPSRC centre for Doctoral Training in Autonomous and Intelligent Machines and Systems EP/S024050/1. MJ is also supported by Amazon Web Services and the Oxford-Singapore Human-Machine Collaboration Initiative. The experiments were made possible by a generous equipment grant from NVIDIA.

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

## A Proof of Theorem 1

Starting from the definition of the momentum term in Adam's update rule:

$$m_t^i = (1 - \beta_1) \sum_{j=-t'}^{t} {\beta_1}^{t-j} g_j^i,$$

$$= (1 - \beta_1) \left[ g \sum_{j=-t'}^{-1} {\beta_1}^{t-j} + kg \sum_{j=0}^{t} {\beta_1}^{t-j} \right],$$

$$= (1 - \beta_1){\beta_1}^t g \left[ \sum_{j=-t'}^{-1} {\beta_1}^{-j} + k \sum_{j=0}^{t} {\beta_1}^{-j} \right],$$

$$= (1 - \beta_1){\beta_1}^t g \left[ \beta_1 \sum_{j=0}^{t'-1} {\beta_1}^{j} + k \sum_{j=0}^{t} ({\beta_1}^{-1})^{j} \right].$$

From the solution to the sum of a geometric series:

$$m_t^i = (1 - \beta_1){\beta_1}^t g \left[ \beta_1 \frac{1 - {\beta_1}^{t'}}{1 - \beta_1} + k \frac{1 - {\beta_1}^{-(t+1)}}{1 - {\beta_1}^{-1}} \right],$$

$$= (1 - \beta_1){\beta_1}^t g \left[ \beta_1 \frac{1 - {\beta_1}^{t'}}{1 - \beta_1} + k \frac{{\beta_1}^{-t} - \beta_1}{1 - \beta_1} \right],$$

$$= g \left[ {\beta_1}^{t+1}(1 - {\beta_1}^{t'}) + k(1 - {\beta_1}^{t+1}) \right].$$

Similarly for $v_t^i$, it follows:

$$v_t = g^2 \left[ {\beta_2}^{t+1}(1 - {\beta_2}^{t'}) + k^2(1 - {\beta_2}^{t+1}) \right].$$

Substituting $v_t^i$ and $m_t^i$ into the Adam momentum updates with $\epsilon = 0$ yields:

$$\frac{\hat{m}_{-t',t}^i}{\sqrt{\hat{v}_{-t',t}^i}} = \frac{\sqrt{1 - {\beta_2}^{t'+t+1}}}{1 - {\beta_1}^{t'+t+1}}$$

$$\cdot \frac{g \left[ {\beta_1}^{t+1}(1 - {\beta_1}^{t'}) + k(1 - {\beta_1}^{t+1}) \right]}{\sqrt{g^2 \left[ {\beta_2}^{t+1}(1 - {\beta_2}^{t'}) + k^2(1 - {\beta_2}^{t+1}) \right]}}.$$

Taking the limit $t' \to \infty$ with $\beta_1, \beta_2 \in [0, 1)$ yields our desired result:

$$\lim_{t' \to \infty} \frac{\hat{m}_{-t',t}^i}{\sqrt{\hat{v}_{-t',t}^i}} = \frac{{\beta_1}^{t+1} + k(1 - {\beta_1}^{t+1})}{\sqrt{{\beta_2}^{t+1} + k^2(1 - {\beta_2}^{t+1})}}.$$

## B Results comparison with Asadi et al.

Asadi et al. [10] find in their paper that their method, when applied to DQN, gives roughly comparable performance to their Adam baseline. However, in our paper we find that Adam-MR performs significantly worse than the Adam baseline, even when compared on the same games as in Figure 6. There Adam-Rel performs better than Adam on the inter-quartile mean, but worse on the median. However, given this is a selection of just 12 games of very different difficulties, the median is often likely in this case to reduce to a single game for most algorithms.

To investigate this disparity, we compare our results for Adam-MR to theirs in Table 1. We estimated their scores in each game from the appropriate figures in their paper. Overall we see that our

Table 1: Comparison with the results from Asadi et al. [10]. The scores are estimated by taking the performance at 40M frames from the figures in their paper. We compare to both $K = 1000$, which is our default hyperparameter, and $K = 8000$, which is their default hyperparameter.

| Environment | Score | | | Normalized Score | | |
|---|---|---|---|---|---|---|
| | Adam-MR (K=1000) [10] | Adam-MR(K=8000) [10] | Adam-Rel (K=1000) | Adam-MR (K=1000) [10] | Adam-MR (K=8000) [10] | Adam-Rel (K=1000) |
| Amidar | 350 | 300 | $270 \pm 20$ | 0.20 | 0.17 | $0.16 \pm 0.01$ |
| Asterix | 3500 | 4200 | $3600 \pm 700$ | 0.39 | 0.48 | $0.40 \pm 0.09$ |
| BeamRider | 3800 | 4300 | $4800 \pm 500$ | 0.21 | 0.24 | $0.27 \pm 0.03$ |
| Breakout | 160 | 200 | $300 \pm 20$ | 5.5 | 6.9 | $10.5 \pm 0.7$ |
| CrazyClimber | 0 | 85000 | $80000 \pm 9000$ | -0.41 | 2.85 | $2.6 \pm 0.3$ |
| DemonAttack | 3300 | 3500 | $8400 \pm 500$ | 1.73 | 1.84 | $4.5 \pm 0.3$ |
| Gopher | 3500 | 4000 | $1500 \pm 300$ | 1.50 | 1.74 | $0.6 \pm 0.1$ |
| Hero | 1500 | 6000 | $1200 \pm 600$ | 0.015 | 0.17 | $0.005 \pm 0.02$ |
| Kangaroo | 10500 | 8250 | $6000 \pm 900$ | 3.5 | 2.75 | $2.0 \pm 0.3$ |
| Phoenix | 4250 | 4500 | $3800 \pm 1000$ | 0.54 | 0.58 | $0.5 \pm 0.2$ |
| Seaquest | 1300 | 6000 | $1800 \pm 300$ | 0.03 | 0.14 | $0.042 \pm 0.006$ |
| Zaxxon | 1000 | 6200 | $2200 \pm 300$ | 0.11 | 0.67 | $0.24 \pm 0.04$ |
| Mean | | | | 1.11 | 1.54 | $1.81 \pm 0.2$ |
| Inter-Quartile Mean[3] | | | | 0.49 | 0.92 | 0.66 |
| Median | | | | 0.30 | 0.63 | 0.43 |

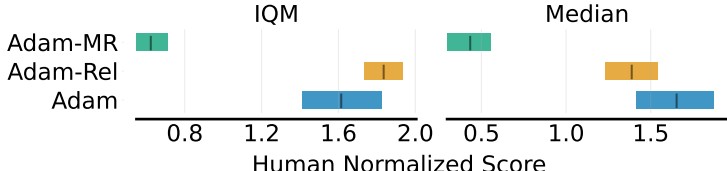

Figure 6: Comparison of the inter-quartile mean and median of Adam-MR, Adam-Rel and Adam on the Atari games evaluated on in Asadi et al. [10].

implementation, which uses $K = 1000$, performs significantly better than their implementation with $K = 1000$. It is also better in mean but worse in median and inter-quartile mean than their $K = 8000$ implementation. In short, our results broadly match theirs reported after a similar amount of training, but our Adam baseline performs significantly better than theirs. However, there are a number of differences in our evaluation. First we run for 10M steps (40M frames) whereas they run for 30M steps (120M frames). Secondly, they use the Dopamine [50] settings for Atari, whereas we use the more standard ones used by DQN [18]. We kept these settings throughout our paper to avoid significant hyperparameter tuning by evaluating in as standard settings as possible. We believe these results demonstrate the correctness of our implementation of their work and that our method still performs favourably.

## C   Results with Polyak Averaging

We also run experiments on DQN with Polyak averaging to examine the effect of Adam-Rel in cases where there are soft-target updates.

We set $\tau = 0.02$. This was chosen so that after 250 steps, the previous target update frequency, the original target parameters would contribute just 0.5% to the new target. We found the best learning rate to be lower, at $5 \times 10^{-5}$. The results are shown in Figure 7. As shown in that figure, Adam-Rel outperforms Adam in this setting as well, achieving a higher median value, although still retaining a long tail of negative results. We also note that although Adam achieves better performance than the baseline without Polyak averaging, Adam-Rel performs worse than when Polyak averaging is not used. This may be due to resetting $t$ being less effective when soft-target changes are used, or that more extensive tuning may improve its performance.

## D   Code Repositories

For the Atari experiments (both DQN and PPO), we based our implementation on CleanRL [19]. This code is available here. For the Craftax experiments, we based our implementation on PureJaxRL [20]. This code is available here.

---

[3]This is the inter-quartile mean over *environments* as opposed to the more usual over *environments and seeds*. This is because Asadi et al. [10] do not provide individual seed data.

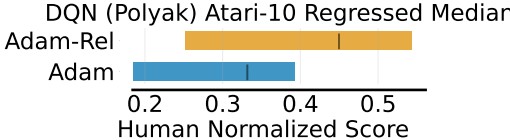

Figure 7: Comparison of the regressed median on the Atari-10 benchmark of Adam and Adam-Rel for DQN with Polyak Averaging.

## E Compute and Additional Experiments

For our DQN experiments, we swept over learning rates for Adam-MR, Adam-Rel and Adam. For PPO experiments, we swept over learning rate, max gradient norm and GAE $\lambda$ values, as we found these to differ from the PPO defaults. Experiments were performed on an internal cluster of NVIDIA V100 GPUs. Experiments were scheduled using slurm, with 10 CPU cores per GPU.

The Atari PPO experiments required around 10000 GPU hours to complete, including hyperparameter tuning. The DQN experiments, because of the computational inefficiency of DQN, take much longer to run (approximately 2 days per experiment), and hence used a total of 14000 GPU hours, despite there being many fewer parallel runs. The Craftax-Classic experiments took around 300 GPU hours to complete.

## F Hyperparameters

Table 2: Atari Adam PPO hyperparameters

| Hyperparameter | Value |
|---:|:---|
| Learning Rate | 0.00025 |
| Number of Epochs | 4 |
| Minibatches | 4 |
| $\gamma$ | 0.99 |
| GAE $\lambda$ | 0.95 |
| Normalise Advantages | True |
| $\epsilon$ | 0.1 |
| Value Function Clipping | True |
| Max Grad Norm | 0.5 |
| Number of Environments | 8 |
| Number of Rollout Steps | 128 |

Table 3: Atari Adam-Rel and Adam-MR PPO hyperparameters

| Hyperparameter | Value |
|---:|:---|
| Learning Rate | 0.002 |
| Number of Epochs | 4 |
| Minibatches | 4 |
| $\gamma$ | 0.99 |
| GAE $\lambda$ | 0.9 |
| Normalise Advantages | True |
| $\epsilon$ | 0.1 |
| Value Function Clipping | True |
| Max Grad Norm | 5.0 |
| Number of Environments | 8 |
| Number of Rollout Steps | 128 |

Table 4: Atari-10 DQN hyperparameters

| Hyperparameter | Value |
| --- | --- |
| Learning Rate | 0.0001 |
| Buffer Size | $1 \times 10^6$ |
| $\gamma$ | 0.99 |
| GAE $\lambda$ | 0.9 |
| Target Network Update Steps | 1000 |
| Batch Size | 32 |
| Start $\epsilon$ | 1 |
| End $\epsilon$ | 0.01 |
| Exploration Fraction | 0.1 |
| Number of Steps without Training | 80000 |
| Train Frequency | 4 |

Table 5: Craftax Adam and Adam-MR PPO hyperparameters

| Hyperparameter | Value |
| --- | --- |
| Learning Rate | 0.0003 |
| Number of Epochs | 4 |
| Minibatches | 4 |
| $\gamma$ | 0.99 |
| GAE $\lambda$ | 0.9 |
| Normalise Advantages | True |
| $\epsilon$ | 0.2 |
| Value Function Clipping | True |
| Max Grad Norm | 1 |
| Number of Environments | 512 |
| Number of Rollout Steps | 64 |

Table 6: Craftax Adam-Rel hyperparameters

| Hyperparameter | Value |
| --- | --- |
| Learning Rate | 0.001 |
| Number of Epochs | 4 |
| Minibatches | 4 |
| $\gamma$ | 0.99 |
| GAE $\lambda$ | 0.7 |
| Normalise Advantages | True |
| $\epsilon$ | 0.2 |
| Value Function Clipping | True |
| Max Grad Norm | 5 |
| Number of Environments | 512 |
| Number of Rollout Steps | 64 |

