# OpenReview forum: "Adam on Local Time: Addressing Nonstationarity in RL with Relative Adam Timesteps"
_NeurIPS.cc/2024/Conference — NeurIPS 2024 poster_

### Official Review · Reviewer_SmBt · 2024-07-06

**Soundness:** 2
**Presentation:** 3
**Contribution:** 2
**Rating:** 4
**Confidence:** 4

**Summary:**

Due to problems with momentum and a non-stationary target, the authors propose resetting the value of ‘t’ in Adam, which is used to determine bias correction on the momentum terms, when updating the target. This approach is validated on PPO and DQN in Atari and Craftax.

**Strengths:**

- Performance benefits over two key algorithms (PPO and DQN) when used with Adam.
- Simple to implement idea.

**Weaknesses:**

- In terms of new insight, this paper offers very little that hasn’t been said in prior work (Bengio et al., Asadi et al.). Similarly, analysis in the paper is based around the update size. However, simply resetting would also bound the update size, so I feel like there is some lacking analysis or evidence to explain why maintaining momentum estimates is a reasonable approach.
- Since the idea is not analyzed in depth, I would say there is limited empirical evidence (in terms of evaluations over environments and algorithms), especially compared to prior work.

**Questions:**

It would be interesting to see more analysis/insight either empirically or theoretically with regards to what happens to the gradient. For example, do momentum estimates correlate with the error terms over new iterations? While resetting seems like a brute force approach to mitigating problems with momentum and non-stationarity, it’s unclear that maintaining the momentum estimate is necessarily better.

**Limitations:**

Satisfactory.

---

> ### Author Rebuttal · Authors · 2024-08-07
>
> Thank you for your review.
>
> Regarding your comment on new insight, we ask where Adam's $t$ parameter is studied in either of the papers your shared, or any prior work? Furthermore, is there any suggestion that resetting this parameter can theoretically bound the update size, or have the empirical impact on performance presented here? If not, what further insight would go beyond the "very little" you suggest we present?
>
> Regarding your suggestion that Adam-MR also bounds the update size, we ask why it is more reasonable to *also* reset the momentum when resetting only $t$ is sufficient to bound the update size? Our evalution thoroughly demonstrates the empirical benefit of this and provides strong evidence for maintaining momentum estimates.
>
> Regarding our method not being analysed in depth, we provide a detailed background on Adam, analyse the update size with and without timestep resetting, and provide a formal proof analysing the impact of sudden changes in update size. Is there a specific element of this analysis you believe to be missing? If not, we do not believe that the analysis being not "in depth" is a well-founded criticism.
>
> Regarding our empirical evidence being limited compared to prior work, we note that, compared to prior work, we evaluate on an additional class of algorithms (on-policy) and a new environment (Craftax), and that all other reviewers praise the thoroughness of our empirical evaluations. Without a concrete suggestion for further experiments or what empirical evidence is missing, this appears to be a generic criticism that could be made on any paper and we ask that you reconsider this point.
>
> In response to your question, we again ask why the momentum estimate should be reset given our demonstration that timestep resetting bounds the update size and our extensive results demonstrating that resetting momentum harms performance.
>
> We hope you will be able to provide further clarification regarding the points raised above. If not, given our reasoning and your praise for our paper, we hope that you will either raise your score or explain the barriers preventing acceptance?

---

> > ### Comment · Reviewer_SmBt · 2024-08-12
> >
> > Thank you for the response.
> >
> > Prior work has shown possible limitations with Adam in the RL setting. Your work builds on this insight. Further insight might include new settings (e.g., EMA), or deeper analysis of the problem.
> >
> > If one of your key arguments for the performance boost is that the proposed approach bounds the update size, then one might expect other approaches which also bound the update size to provide a similar benefit. Since this appears not to be the case, it suggests that you have not correctly, or fully, identified the problem that your method corrects.
> >
> > The analysis is not in-depth because only one element of the algorithm is considered is detail (update size). However, analysis on update size is only sufficiently valuable if we are certain that update size is the problem. I don't see why this is true.
> >
> > Regarding empirical evidence, while I appreciate the current set of experiments, I do not believe that they are thorough enough given the limited insight. If the other reviewers feel otherwise, then I simply disagree with the other reviewers. 2 algorithms and 2 domains does not convince me this is a general-purpose RL technique.
> >
> > This paper either needs more convincing experiments, empirical insights, or stronger theoretical results to pass the bar for acceptance (one of three, not necessarily all three). In its current state, I view this as a minor heuristic change to prior work with insufficient justification. As such, I do not believe the paper is above the bar for acceptance.

---

> > > ### Author Response · Authors · 2024-08-12
> > >
> > > Thank you for taking the time to respond to our rebuttal. There appear to be two concerns with our work, which we address below.
> > >
> > > Firstly, you mentioned a concern with the experiments conducted, saying that you do not believe that they are “thorough enough” and that “2 algorithms and 2 domains does not convince me this is a general-purpose RL technique.” In this respect, we believe that precedent from past submissions provides the fairest guide for what merits acceptance. Previous similar empirical research [1,3,4,5,6] from top conferences has focussed on only one algorithmic setting and evaluated on a single environment, typically Atari. Furthermore, both of the works that you cite [1,2] do not have as thorough empirical evaluations, either not evaluating in multiple domains or in on-policy and off-policy settings. Whilst you are entitled to hold higher standards for empirical rigor, we exceed the standard for empirical evaluation set by past top conferences, and hope you will reconsider your decision on these terms.
> > >
> > > Secondly, you mention that “one might expect other approaches which also bound the update size to provide a similar benefit. Since this appears not to be the case, it suggests that you have not correctly, or fully, identified the problem…” We entirely agree with your point that other methods also bound update size and that update size is not the full extent of the optimization problem. Where we disagree is that optimization in RL is multifaceted and there are a number of issues to consider, some of which alternative methods sacrifice. As a reductive example, an optimizer which never updated the agent would bound update size, but would not provide similar benefit to our method. Instead, treating update size as one of many well-established considerations in optimization [7, 8], we theoretically demonstrate our method’s impact in this aspect, then empirically validate its impact on overall RL performance. As there is no single issue and no theoretical or empirical result can entirely encapsulate the challenges in optimization, we hope you agree that this is not a fair standard for a paper.
> > >
> > > Finally we would like to draw your attention to some new results we have obtained, evaluating our setting when Polyak averaging is used.
> > >
> > > We ran additional experiments in the Atari-10 setting, evaluating DQN with Adam and Polyak averaging against DQN with Adam-Rel and Polyak averaging. We cannot upload the results figure here, but we show the results below, where the values in brackets are the lower and upper 95% bootstrapped confidence intervals respectively.
> > >
> > > | Method                         | Regressed Median    |
> > > |--------------------------------|---------------------|
> > > | Adam (w/ Polyak Averaging)     | 0.096 (0.089, 0.13) |
> > > | Adam-Rel (w/ Polyak Averaging) | 0.42 (0.26, 0.50)   |
> > >
> > > This also demonstrates that our method extends to the case of Polyak averaging.
> > >
> > > These results got a slightly lower average score -- Polyak averaging is not typically used in Atari with DQN. We chose the coefficient, $\tau$, to result in a similar length of optimisation to the original hyperparameters. We therefore set $\tau = 0.98$ so that after 250 updates, the original parameters contribute $0.6\%$ to the target parameters. Although the scores attained are lower, this change seems to affect Adam and Adam-Rel similarly.
> > >
> > > We hope that our response, in addition to these extra experimental results, encourages you to raise your score to vote for acceptance.
> > >
> > > [1] Asadi, Kavosh, Rasool Fakoor, and Shoham Sabach. "Resetting the optimizer in deep rl: An empirical study." Advances in Neural Information Processing Systems 36 (2024).
> > >
> > > [2] Bengio, Emmanuel, Joelle Pineau, and Doina Precup. "Correcting momentum in temporal difference learning." arXiv preprint arXiv:2106.03955 (2021).
> > >
> > > [3] Lyle, Clare, et al. "Understanding plasticity in neural networks." International Conference on Machine Learning. PMLR, 2023.
> > >
> > > [4] Schwarzer, Max, et al. "Bigger, better, faster: Human-level atari with human-level efficiency." International Conference on Machine Learning. PMLR, 2023.
> > >
> > > [5] Ceron, Johan Samir Obando, Marc G. Bellemare, and Pablo Samuel Castro. "Small batch deep reinforcement learning." Thirty-seventh Conference on Neural Information Processing Systems. 2023.
> > >
> > > [6] Lyle, Clare, Mark Rowland, and Will Dabney. "Understanding and Preventing Capacity Loss in Reinforcement Learning." International Conference on Learning Representations.
> > >
> > > [7] Goyal, Priya, et al. "Accurate, large minibatch sgd: Training imagenet in 1 hour." arXiv preprint arXiv:1706.02677 (2017).
> > >
> > > [8] Gotmare, Akhilesh, et al. "A closer look at deep learning heuristics: Learning rate restarts, warmup and distillation." arXiv preprint arXiv:1810.13243 (2018).

---

### Official Review · Reviewer_37TD · 2024-07-09

**Soundness:** 4
**Presentation:** 3
**Contribution:** 3
**Rating:** 6
**Confidence:** 4

**Summary:**

One of the main challenges of reinforcement learning is its inherent nonstationary nature. Such non-stationarity can cause learning difficulties. The tools currently available for deep reinforcement learning are largely borrowed from deep learning, such as the Adam optimizer, which this paper focuses on. The authors show that Adam, under nonstationarity, can have large updates leading to learning difficulties. Thus, they propose a simple modification to Adam in which they reset the time parameter at every epoch in PPO or every target update in DQN. This slight modification seems to significantly improve performance, agreeing with the observations by several works that proposed similar modifications to Adam [1,2,3] and showed gains in performance.
\
\
[1] Emmanuel Bengio, Joelle Pineau, and Doina Precup. Correcting momentum in temporal difference learning. arXiv preprint arXiv:2106.03955, 2021.
\
[2] Kavosh Asadi, Rasool Fakoor, and Shoham Sabach. Resetting the optimizer in deep rl: An empirical study. arXiv preprint arXiv:2306.17833, 2023.
\
[3] Shibhansh Dohare, Qingfeng Lan, and A Rupam Mahmood. Overcoming policy collapse in deep reinforcement learning. In Sixteenth European Workshop on Reinforcement Learning, 2023.

**Strengths:**

The authors present a novel and simple approach to solving a big problem. This is especially promising since it will allow quick wide adoption in the future. The authors introduced the problem carefully and talked about it in such a clear way that guides the reader step by step, and they also provide some theoretical justification for why their method would work from first principles. The evaluation benchmarks seem extensive, suggesting that the methods can be applicable to a wide range of problems.

**Weaknesses:**

- The main weakness of the paper is its limited comparison to other methods, particularly the methods that are closely related to Adam-Rel (e.g. [1,3]). Additionally, the only competitor is Adam-MR [2] which almost always gives worse performance, contradicting previous works. There is no similar theoretical analysis on Adam-MR explaining the poor performance or some empirical evidence convincing the reader why it fails.
- Writing can be improved, especially when explaining parts of Adam. This can be done using more proper mathematical terms such as “estimators” or “correcting the biasedness of an estimator,” etc.
- The title is written in such a way that suggests your method is applicable to a wide range of RL methods, but it seems that this is not the case, and you only focus on PPO and DQN. I suggest a change that reflects your contributions without overstating them.
- No pseudocode is given for Adam-Rel with DQN. It’s inferred from the context that a time parameter reset is done before each target function update, but the pseudocode needs to be there to confirm this for the reader.
- Dohare et al. (2023) showed that policy collapse in PPO with MuJoCo is more pronounced than DQN in Atari. It might be because the observations are bounded in Atari but not in MuJoCo, which increases the non-stationarity. The authors didn’t consider any environments other than pixel-based ones with bounded observation ranges. I think it’s necessary also to show some results on MuJoCo environemnts (or similar environments) to confirm that the results are qualitatively similar in those settings.
\
\
\
**Minor issues:**
- “It uses a learned critic trained by a TD loss to estimate the value function, and a clipped actor update of the form.” -> This is not accurate. PPO doesn’t use a learned critic. The critic is learned as well.
- “over which the above update is calculated” -> You wrote an objective in Eq. 1, not an update
- “Adam, the most popular optimizer that  uses momentum” -> It doesn’t use the momentum you defined in the background. You might want to clear this up for the reader.
\
\
[1] Emmanuel Bengio, Joelle Pineau, and Doina Precup. Correcting momentum in temporal difference learning. arXiv preprint arXiv:2106.03955, 2021.
\
[2] Kavosh Asadi, Rasool Fakoor, and Shoham Sabach. Resetting the optimizer in deep rl: An empirical study. arXiv preprint arXiv:2306.17833, 2023.
\
[3] Shibhansh Dohare, Qingfeng Lan, and A Rupam Mahmood. Overcoming policy collapse in deep reinforcement learning. In Sixteenth European Workshop on Reinforcement Learning, 2023.

**Questions:**

- One of the main assumptions in the theoretical work is that the gradients become near zero at convergence, and then we get sudden large updates once the gradients start to be large. This might be the case for PPO, but it doesn’t seem to be the case for DQN. Dohare et al. (2023) showed that setting beta1 to equal beta improves PPO performance but has a smaller impact on DQN. Adam-Rel shows improvement in both PPO and DQN, but DQN's motivation is not clear. Can the authors explain why they expect Adam-Rel to improve DQN?
- I’m puzzled by Adam-MR's poor performance. Your results seem to contradict (as you mentioned) two previous works. Would it be possible that you used an incorrect implementation for Adam-MR?
- Do the authors have any intuition as to why Adam-MR performs poorly? In particular, do they have a similar theoretical analysis for Adam-MR showing that its updates can be large or even larger than Adam?

**Limitations:**

The authors adequately discussed the limitations of their work.

---

> ### Author Rebuttal · Authors · 2024-08-07
>
> Thank you for your review of our paper. We're glad you find our method to be "novel and simple" and capable of "wide adoption in the future". We also appreciate your comments about the clarity of our writing and theoretical justification, in addition to our evaluation being "extensive". We respond to each of your weaknesses in order below:
>
>
>
> * Regarding your request for further baseline methods, we note that this topic is relatively underexplored and lacks significant prior work. However, despite the work being from a workshop and not yet published at an archival conference, we have now added the method decribed in Dohare et al. [3] to our evaluation on Craftax, which shows that it does not significantly improve performance over Adam and underperforms our method. We will add this baseline to the remaining experiments in the camera-ready copy of this paper.
> Regarding your suggestion that our Adam-MR scores contradict prior work, we refer you to Appendix B of our paper, which explains this discrepancy in detail. In summary, we replicate Adam-MR's original scores in Atari, but evaluate against a significantly stronger baseline. As you state, our experiments are extensive and should be sufficient to convince the reader that Adam-MR fails in the settings studied in our work.
> * We will endeavour to be increasingly formal in future revisions of this work, however we note that our current explanation of Adam is already extensive at two pages and that most reviewers praise the paper's writing and clarity.
> * We evaluate on DQN and PPO, the two predominant algorithms in on-policy and off-policy RL, which you agree suggests the method will be "applicable to a wide range of problems". Therefore, we believe that our results already demonstrate that Adam-Rel will be applicable to a wide range of RL methods and are unsure which additional methods you believe would enable us to make this claim.
> * We are confident that readers can infer the function of Adam-Rel on DQN without complete pseudocode due to the extreme simplicity of our method, as was agreed by all reviewers. You correctly infer the method in your review, however, we would be willing to add this to the appendix if you believe there is ambiguity as to where $t$ is reset.
> * Firstly, regarding your suggestion that we consider only pixel-based environments, we note that Craftax is symbolic and not pixel-based. Secondly, your suggestion that unbounded observations cause policy collapse is interesting, however, it is one of many valid hypotheses for this result. It is equally possible, if not more likely, that differences in the algorithms or other variations between the two environments cause this observation. Since this hypothesis does not appear to have strong support or context, we do not believe it is fair to require its investigation as a barrier to acceptance for this work.
>
> Thank you for suggesting edits in the minor issues. We believe these are inferable from context -- the critic is "learned" as part of the algorithm, the update is generally the derivative of the objective, and we state that we provide only the "typical" formulation of momentum -- so we hope that these do not influence your score. However, we will take care to find less ambiguous wording in the next revision.
>
> Finally, we respond to each of your questions below:
> * We believe this phenomenon would be equally, if not more, pronounced in DQN as the value function undergoes more gradient steps before it is updated. However, verifying this is would provide an interesting area for analysis in future work.
> * We again refer you to Appendix B of our paper, which explains this discrepancy in detail. In summary, we replicate Adam-MR's original scores in Atari, but evaluate against a significantly stronger baseline. We also investigate Adam-MR on an entirely new algorithm (PPO) in two settings, where the performance drop is most pronounced.
> * We believe the Adam-MR results are unsuprising for PPO, as the algorithm has a small number of updates before the momentum is reset. This prevents a reasonable momentum value from being estimated by Adam-MR before it is reset, which predictably hurts performance. Given we demonstrate that resetting $t$ bounds update size, we believe a more pertinent question is why it would be necessary to reset the momentum at all?
>
> Thank you again for taking the time to review our work. To summarise the key rebuttal: we have added an experiment with [3] but do not believe there is other significant prior work; we and most reviewers find our writing clear; our experiments already demonstrate wide applicability by covering the predominant algorithms in on-policy and off-policy RL; we and all reviewers find our method to be simple to understand; we clarified our environments and do not believe that unbounded observations are a necessary hypothesis to investigate for acceptance.
>
> Given this, we as unsure of what further steps we can take to improve the paper based on your feedback. In light of our reasoning, your praise of the method, and your strong scores for Soundness (4), Presentation (3), and Contribution (3), we hope you will consider raising your score to an acceptance.

---

> ### Comment · Reviewer_37TD · 2024-08-11
> **Thank you for your response**
>
> I appreciate the extensive response to my concerns. I thank the authors for adding the comparison with Dohare et al. (2023). It is clear that Adam-Rel performs better than setting $\beta_1=\beta_2$. Given the new result with Dohare et al. (2023) and the authors' willingness to clarify the missing details, including adding the DQN-Adam-Rel pseudocode and fixing other writing issues, I have raised my score.

---

### Official Review · Reviewer_uFqs · 2024-07-11

**Soundness:** 3
**Presentation:** 3
**Contribution:** 2
**Rating:** 5
**Confidence:** 4

**Summary:**

This paper introduces a simple approach to address the issue of large updates commonly encountered with Adam optimizers in deep learning applications. The authors focus on a specific scenario prevalent in deep reinforcement learning: the updating of target networks. Instead of the conventional approach of resetting both the timestep and the momentum variables in the Adam optimizer, the authors propose only resetting the timestep. This modification leads to enhanced learning performance, as demonstrated through evaluations conducted in both the Atari and Craftax environments, utilizing a range of on-policy and off-policy algorithms.

**Strengths:**

The paper is generally clear, particularly benefiting from a detailed explanation of the Adam optimizer’s mechanics, which enhances accessibility for those who may be unfamiliar with the specifics. The simplicity of the proposed modification—only resetting the time step—is a significant advantage, allowing easy implementation and thus potentially reducing errors in implementation. The experiments are thoughtfully designed, aligning closely with the research questions and conducted in the Atari and Craftax environments, which are known for their complexity. Additionally, the selection of baselines, including the standard and a modified version of Adam, demonstrates the proposed approach’s effectiveness against established methods.

**Weaknesses:**

The research question addressed by the authors, focusing solely on large gradient updates due to changes in the target network, seems somewhat narrow. This focus may overlook the broader applicability and efficacy of the proposed solution across various reinforcement learning scenarios. Notably, many algorithms have achieved better stability using softer target updates, like Polyak averaging, which the authors acknowledge [1,2,3]. It would be beneficial to see how the proposed optimizer modification compares in environments where such techniques are traditionally employed.

Moreover, the approach of resetting parameters in an optimizer, though straightforward, could be considered a drastic and potentially inelegant solution. This method might introduce other issues, such as affecting the convergence behavior of the optimizer. While I currently lack an alternative suggestion, exploring methods that adjust the scale of parameter updates dynamically, rather than resetting them, might offer a more refined solution.

[1] Lillicrap, Timothy P., et al. "Continuous control with deep reinforcement learning." arXiv preprint arXiv:1509.02971(2015).

[2] Kaplanis, Christos, Murray Shanahan, and Claudia Clopath. "Continual reinforcement learning with complex synapses." International Conference on Machine Learning. PMLR, 2018.

[3] Schwarzer, Max, et al. "Bigger, better, faster: Human-level atari with human-level efficiency." International Conference on Machine Learning. PMLR, 2023.

**Questions:**

1. Have the authors considered extending the setting towards changes in data distribution rather than only due to a change in target network?
2. The paper includes a comprehensive study using both off-policy and on-policy algorithms. Could the authors clarify the rationale behind this choice? Specifically, which type of algorithm is more affected by drastic changes in data distribution, and why?
3. Figure 4 and Figure 5 should be re-labelled such that Figure 4 should be Figure 5 and Figure 5 should be Figure 4 since in the current version, Figure 5 appears before Figure 4.
4. I find Figure 5 hard to interpret. Could the authors provide more detail on what the curves represent? Additionally, enhancing the clarity of the caption might help.
5. Line 86: The authors mention a change of objective. However, it seems the objective function remains consistent throughout. Could the authors clarify if they are actually referring to changes in data distribution rather than the objective itself?
6. Line 259: The authors mention 'matching the trend.' Do you mean matching the overall trend rather than the specific shape or pattern?
7. Line 261:  Could the authors clarify what you mean by 'overshooting' and 'undershooting'? The current usage is confusing.
8. Line 283: The use of 'However' at the beginning of Line 283 immediately follows another sentence starting with 'However' in Line 280. Could these be rephrased to improve the flow of the text?

**Limitations:**

None. The authors have presented the concerned limitations of their work in the paper.

---

> ### Author Rebuttal · Authors · 2024-08-07
>
> Thank you for taking the time to review our paper. We were pleased with your extensive praise of our paper, finding the writing "generally clear, particularly benefitting from a detailed explanation of the Adam optimizer's mechanics", the "simplicity of the method a significant advantage", the experiments "thoughtfully designed, aligning closely with the research directions and conducted in...environments, which are known for their complexity", and that the baselines "demonstrate the proposed approach's effectiveness against other methods".
>
> Your first criticism of our paper is that the setting seems "somewhat narrow". We strongly disagree with this characterization. We demonstrate results in the most studied on-policy and off-policy RL algorithms. No other reviewer has raised this as a concern, with all other reviewers praising the applicability of our approach.
>
> Secondly, you criticize our solution as "drastic and potentially inelegant", further stating that it "might introduce other issues, such as affecting the convergence behavior of the optimizer", and finally suggesting we try to find a more "refined" solution. Firstly, this criticism contradicts *all reviewers* who praise our approach's simplicity. Secondly, the claim that our approach "might introduce other issues", without a clear explanation as to how or why those issues might occur is entirely generic. Such generic claims are unfair and invalid criticism.
>
> Thirdly, while we could always do more theoretical analysis on the convergence properties of our approach, we provide extensive empirical results that demonstrate our approach's ability to find higher return policies than relevant baselines across on-policy and off-policy RL. Finally, we find your generic criticism of our work using imprecise terms such as "drastic" and "inelegant" entirely unfair.
>
> We respond to your questions one by one below.
>
> 1. We do not consider changes in data distribution in this work. While we think this is an interesting area of research, we believe that methods to address such problems are likely to look different to our approach and are best studied in future work. We believe our current contribution stands well enough alone without such additions.
> 2. As discussed above, we do not address data distribution shifts in this work. For target nonstationarity, we would expect that value-based methods are more affected by this given their long optimisation times with fixed target networks. Our method's stronger performance in DQN hints at this, but more research is likely needed to verify this hypothesis.
> 3. Thank you for pointing this out. We have fixed it in our copy and will include this in the camera ready.
> 4. In PPO, you collect new data and optimize a stationary objective, before repeating the process with your new actor and critic. Each optimization is thus over a stationary objective. In Figure 5, we average over these stationary objectives throughout training and analyze the gradient (i.e. before the optimizer) and update (i.e. after the optimizer) properties. This rebuttal is too short for a detailed discussion of the findings, for which we refer you to the paper.
> 5. In the section following line 86, we clearly and precisely explain our setting, clarifying on line 90 that the $\phi$ used in $L(\theta, \phi)$ does not include the data distribution. Would you be able to clarify your misunderstanding here? We clearly state in that section we do not investigate the data distribution.
> 6. Yes exactly, we mean that the overall shape matches.
> 7. By overshooting and undershooting, we mean increasing (or decreasing respectively) over a threshold. We will amend this section in a camera-ready version to clarify this.
> 8. Thank you for pointing this out, we will amend it in a camera-ready version.
>
> Thank you again for taking the time to review our paper. We were surprised that, given your extensive praise of our paper and your listed weaknesses, your final rating was a reject. We hope that our reasoning here encourages you to raise your score to a rating that reflects all points made in your review.

---

> > ### Comment · Reviewer_uFqs · 2024-08-07
> > **Rebuttal Follow up**
> >
> > Thank you for your rebuttal. Upon reading your comments, I first want to apologize for any confusion my review may have caused. It was never my intention to undermine the authors’ work. More importantly, I recognize that I should provide more detailed insights and reasoning to support the claims I made. Let me unpack my claims further, and I would be glad to clarify if needed.
> >
> > # 1. Narrow Setting
> > The scenario referred to as narrow pertains to the situation the authors address: a (hard) change in the target network. The authors claimed (lines 10-11) and demonstrated that changes in the target network impact learning performance, as shown in Figure 2.
> >
> > The reason for my claim that the study is narrow is as follows: Do learning difficulties persist if we use a soft target network update, such as Polyak averaging, instead of a hard update?
> >
> > Do we still observe large updates? If Polyak averaging can overcome learning inefficiencies and the issue of large updates, what additional benefits does the proposed method, Adam-Rel, provide, given that Polyak averaging offers a simpler solution without resetting?
> >
> > # 2. Why is Resetting Drastic?
> > Resetting is drastic because it generally removes previously learned information, such as momentum, and affects adaptive learning rates, which can result in longer convergence times.
> >
> > # 3. Why is Resetting potentially inelegant?
> > Resetting disrupts the continuous learning process, causing the model to lose valuable accumulated optimization information.
> >
> > # 4. What would be a more refined solution?
> > A potentially more refined solution would be to dynamically adapt Adam's timestep, such as using the difference in the gradient update norm. This approach would be less drastic and more elegant.
> >
> > # Q2: Impact of Data Distribution Change
> > I refer to the change in data distribution after a hard update of the network generated by a different policy. How does this affect off-policy algorithms like DQN and on-policy algorithms like PPO? Are they both impacted in the same way? If not, which class of algorithms experiences more drastic effects and why?
> >
> > # Q4: Suggestions for Figure 5
> > Label each subfigure in Figure 5, resulting in 5a, 5b, 5c, and 5d for easier reference.
> > Why is the trend in the update norm of Adam so different compared to the theoretical model?
> > The gradient norm per parameter update for both Adam and Adam-Rel exhibits the same trend, which is vastly different from the theoretical model. Why is this so?
> >
> > To the best of my knowledge, these effects were not discussed in the main paper.
> >
> > # Q5: Clarification on Parameters
> > From lines 87-90, you state: “More explicitly, we consider optimizing a stationary loss function $L(\theta, \phi)$, where $\theta$ are the parameters to be optimized and $\phi$ is the other parameters of the loss function (such as the parameters of a value network), which are not updated throughout optimization, but does not include the training data.”
> > What do the parameters $\theta$ consist of? Are they the parameters of the Adam optimizer or the parameters of the ANN pr both? Why is the value network not being updated during optimization?
> >
> > The phrase “which are not updated throughout optimization, but does not include training data” is confusing. Does this imply that the training data is updated throughout optimization?
> >
> > # Conclusion
> >
> > My main reason for rejecting this submission is due to my concerns about the limited and narrow setting the authors have studied. I have reviewed the feedback from other reviewers, and I am still not convinced that this work warrants acceptance. However, I will re-evaluate based on the progress of the discussion between the authors and the other reviewers, as well as our ongoing discussion.

---

> > > ### Author Response · Authors · 2024-08-08
> > > **Thank you for the Response**
> > >
> > > Thank you very much for the clarification of your review and extensive response to our comments. We address your concerns below.
> > >
> > > ## The setting
> > >
> > > Your objection to our work, as we understand it, is that you are concerned that polyak averaging, which we left out discussion of, would address precisely the problem that we are trying to solve.
> > >
> > > However, this is not the point. There may be many solutions to the phenomenon that we describe. Our goal here is not to present **the only solution** but instead to **investigate current common practice and propose a simple and practical solution**. The methods we investigate are popular and widely used and therefore relevant for our paper. Our solution of resetting the timestep has a number of advantages:
> > > 1. It can be applied to both on-policy and off-policy methods
> > > 2. It is very simple to implement
> > > 3. It has strong empirical performance in a range of environments
> > >
> > > Polyak averaging, although it satisfies the latter two points, cannot be applied to PPO, or related algorithms, as there is no target in a similar way. Therefore, even if
> > > Polyak averaging did indeed address this problem, it's still not clear it is a better solution than what we have proposed.
> > >
> > > Thank you for clarifying your remarks about our method being
> > > "drastic" and "potentially inelegant". On our method being "drastic", you state:
> > >
> > > > Resetting is drastic because it generally removes previously learned information, such as momentum, and affects adaptive learning rates, which can result in longer convergence times.
> > >
> > > We are not sure what you mean by this. Our method **does not remove momentum information**. Adam-Rel retains momentum information. It is clear that this is essential to the performance of our method given the poor performance of Adam-MR on the same tasks. Throwing away momentum information seems to significantly harm performance. Secondly, although our method clearly does affect adaptive update sizes, we **used exactly the same learning rate schedules as the base implementation**. Therefore, although this could potentially cause problems with our method, empirically, using identical learning rate schedules as with Adam achieved similarly good results.
> > >
> > > You also state
> > >
> > > > Resetting disrupts the continuous learning process, causing the model to lose valuable accumulated optimization information.
> > >
> > > **This is not true.** No information is 'lost' by resetting. We merely reset the $t$ parameter to $0$, which is only used to scale the momentum terms. The momentum estimates are exactly the same after as they were before. Instead, resetting $t$ rescales the update, but still retains all information that was there previously.
> > >
> > > ## Data Distribution
> > >
> > > The only way to change the data distribution is to update the policy you roll out to collect data. In this sense,
> > > a change in target, either by updating $\pi$ in PPO or by updating the target network, do not have any effect on this.
> > >
> > > Therefore the only consideration here is how much individual policy updates change the encountered data distribution. In this sense, the *mechanism* that causes this problem is the same. However, this is not a well-studied phenomenon and thus a definitive answer is not possible. If I were to guess, I would think that PPO is less-affected by this problem than DQN because of its clipped objective. However, this is entirely conjecture.
> > >
> > > ## Parameters
> > >
> > > By $\theta$ we just mean the parameters of any neural networks that are optimised. This includes the critic, if it exists, and the policy. Thank you for pointing out this source of confusion -- we will amend this wording in the camera ready copy.
> > >
> > >
> > > Thank you again for engaging with our rebuttal! We hope that the above clarifications encourage you to raise your review score.

---

> > > > ### Author Response · Authors · 2024-08-12
> > > > **Discussion period is closing**
> > > >
> > > > Thank you for your engagement with our previous response. The discussion period is closing soon and we would like to remind you to respond to our previous comment, as well as inform you about some new results we have obtained.
> > > >
> > > > To check your hypothesis about polyak averaging, we ran additional experiments in the Atari-10 setting, evaluating DQN with Adam and polyak averaging against DQN with Adam-Rel and polyak averaging. We cannot upload the results figure here, but we show the results below, where the values in brackets are the lower and upper 95% bootstrapped confidence intervals respectively.
> > > >
> > > > | Method                         | Regressed Median    |
> > > > |--------------------------------|---------------------|
> > > > | Adam (w/ Polyak Averaging)     | 0.096 (0.089, 0.13) |
> > > > | Adam-Rel (w/ Polyak Averaging) | 0.42 (0.26, 0.50)   |
> > > >
> > > > This also demonstrates that our method extends to the case of Polyak averaging.
> > > >
> > > > These results got a slightly lower average score -- polyak averaging is not typically used in Atari with DQN. We chose the coefficient, $\tau$, to result in a similar length of optimisation to the original hyperparameters. We therefore set $\tau = 0.98$ so that after 250 updates, the original parameters contribute $0.6\%$ to the target parameters. Although the scores attained are lower, this change seems to affect Adam and Adam-Rel similarly.
> > > >
> > > > We hope that these additional results, along with our rebuttal and other positive qualities of our paper, convince you to raise your score.

---

> > > > > ### Comment · Reviewer_uFqs · 2024-08-12
> > > > > **Thank you for your response**
> > > > >
> > > > > I thank the authors for their response and providing additional empirical evidence. I believe my concerns have been sufficiently addressed so I will increase my score.

---

### Official Review · Reviewer_JkaJ · 2024-07-15

**Soundness:** 3
**Presentation:** 2
**Contribution:** 3
**Rating:** 6
**Confidence:** 4

**Summary:**

This paper studies the effect of non-stationarity on the Adam optimizer. It shows that the standard use of the Adam optimizer can lead to large updates, which can cause sub-optimal performance. To address this issue, Adam-Rel is introduced, which resets Adam's timestep parameter to zero when the target network changes. Finally, experiments show that Adam-Rel provides performance improvements over Adam.

**Strengths:**

The paper is generally well-written. The paper studies an important problem of large updates caused by Adam in non-stationary settings like reinforcement learning. Explicitly studying basic components of modern deep-RL, like Adam, in non-stationary settings is an important direction of research. The proposed solution, Adam-Rel, is simple and easy to implement.

**Weaknesses:**

The same problem of large updates by Adam in non-stationary problems has been studied before (Lyle et al., 2023; Dohare et al., 2023). They both use the solution of setting $\beta_1 = \beta_2$. The authors seem aware of this proposed solution because the work by Dohare et al. (2023) is discussed in the paper. However, Adam-rel is not compared with Adam with $\beta_1 = \beta_2$. A comparison with Adam with $\beta_1 = \beta_2$ is needed to accept this paper. Even if Adam with $\beta_1 = \beta_2$ performs better, this paper can be accepted, as it studies the problem in much more detail than any prior work.

I'm willing to change my score if this comparison is added to the paper.


Lyle et al. Understanding plasticity in neural networks. ICML, 2023.
Dohare et la. (2023). Overcoming policy collapse in Deep RL. EWRL 2023.

**Questions:**

How does Adam with $\beta_1 = \beta_2$ perform in your experiments?

**Limitations:**

The authors adequately discuss the limitations.

-----------------
UPDATE: I increased my score as the authors provide comparison with Dohare et al. (2023).

---

> ### Author Rebuttal · Authors · 2024-08-07
>
> Thank you for your review of our work. We are pleased that you found the paper "generally well-written", studying an "important problem" and the proposed solution "simple and easy to implement".
> We direct you to the common response for a summary of the issues raised by reviewers and our response to them.
>
> Your sole issue with the paper is that we do not include a comparison with the work of Dohare et al. [1]. Although it was not possible given time constraints to rerun all the experiments, we evaluated $\beta_1 = \beta_2$ in Craftax and found it achieves broadly similar performance as Adam, underperforming our method. See the uploaded PDF for details.
>
> Despite this, we were also surprised that you have initially rejected our paper, especially given that you state
>
> > Even if Adam with $\beta_1 = \beta_2$ performs better, this paper can be accepted, as it studies the problem in much more detail than any prior work.
>
> If the paper can be accepted regardless of the result of this experiment, then the paper should also be accepted without it, albeit perhaps with a lower score. We hope that this reasoning and the new results will encourage you to raise your score significantly.

---

> > ### Comment · Reviewer_JkaJ · 2024-08-11
> >
> > Thank you for your response and for providing a comparison with Adam $\beta_1 = \beta_2$. The new results address my concerns, and I've raised my score to reflect that.

---

### Author Rebuttal · Authors · 2024-08-07

We summarise the strengths and weaknesses from all reviews below.

## Strengths

**Writing** Reviewer **JkaJ** praises the paper as "generally well-written", whilst Reviewers **uFqs** and **37TD** praise the paper's clarity, respectively commenting that the paper "benefits from a detailed explanation of the Adam optimizer's mechanics" and is presented in "a clear way that guides the reader step-by-step".

**Importance** Reviewers **JkaJ**, **uFqs**, and **37TD** praise the importance of the problem setting, respectively describing it as "an important direction of research", "commonly encountered with Adam optimizers in deep learning", and "a big problem...it will allow quick wide adpotion in the future", whilst Reviewer **SmBt** notes the method's "performance benefits over two key algorithms".

**Method** All reviewers (**JkaJ**, **uFqs**, **37TD**, **SmBt**) praise the simplicity and ease of implementation of the method.

**Results** Reviewers **JkaJ**, **uFqs**, and **37TD** praise the design and scope of our evaluation, respectively describing it as "studying [Adam with non-stationarity] in much more detail that any prior work", "thoughtfully designed...conducted in [environments] which are known for their complexity", and "extensive, suggesting that [Adam-Rel is] applicable to a wide range of problems".

## Weaknesses


**Writing** Reviewer **37TD** suggests that the writing "can be improved, especially when explaining parts of Adam." Given the current explaination is over two pages, we would struggle to fit further detail in the main body, but will endeavour to do so in the appendix. We also note that Reviewer **uFqs** praises the "detailed explanation of the Adam optimizer's mechanics".

**Importance** Reviewer **uFqs** complains that our problem setting is "somewhat narrow" because we only investigate instant target changes. We provide experiments that demonstrate its effectiveness in DQN and PPO, the two most studied deep RL methods encompassing both on-policy and off-policy RL. Additionally, no other reviewer raised this as a concern, with all other reviewers the applicability of our approach.



**Method** Reviewer **uFqs** also complains that our method could be considered a "drastic and potentially inelegant solution", suggesting that it "might cause issues with the convergence of the optimizer" and that we should find "a more refined solution". This is an unscientific criticism for which we provide no response. Suggesting an unspecified negative impact and using undefined terms such as "inelegant", "drastic" and "refined" to assess the quality of work has no place in peer review.

**Evaluation** Reviewer **37TD** suggests that Adam-MR's poor performance contradicts previous works, and asks whether we have an incorrect implementation. However, we explain this discrepancy in detail in Appendix B. To summarise this, we replicate the scores of Asadi et al. [1] in Atari, but evaluate against a significantly stronger baseline.

Reviewer **37TD** also cites Dohare et al.'s result that DQN in Atari has less performance collapse than PPO in MuJoCo, and suggests that we should evaluate in a setting that is not pixel based and without bounded observations. Firstly, the claim that we only evaluate in pixel-based settings is incorrect as Craftax has a symbolic observation space. Secondly, whilst bounded observations are one possible hypothesis for this result, there are a plethora of alternative explanations. This previous result could equally be explained by PPO's learning rate schedule, or other differences between these algorithms. Therefore, it is unscientific to suggest that these experiments be a barrier to acceptance given their lack of grounding.

Finally, Reviewer **37TD** criticises our title's claim of applying to a wide range of RL methods. We evaluate on DQN and PPO, the two predominant algorithms in on-policy and off-policy RL, and the same reviewer states that our method will have "quick wide adoption in the future" and be "applicable to a wide range of problems". As the reviewer suggests no additional algorithm that might offer further insight, it is unclear what might allow us to make this claim.

Reviewer **SmBt** claims that there is "limited emprical evidence...particularly compared to prior work". Unfortunately, they provide no suggestion as to what future experiments could be added, making the comment a generic criticism. Compared to prior work, we evaluate on an additional class of algorithms (on-policy) and a new environment (Craftax), and note that all other reviewers praise the thoroughness of our empirical evaluations.

**Prior Work** Reviewers **JkaJ** and **37TD** request comparison with other prior work, in particular Dohare et al. [1], an unpublished workshop paper. While it is not possible given the limited time to rerun all experiments with this baseline, **we add an evaluation of [1] in Craftax to the PDF**, which shows that it does not significantly improve performance over Adam and underperforms our method. In a camera-ready copy of the paper, we will add this baseline to all experiments.

Reviewer **SmBt** claims that our work "offers very little that hasn't been said in prior work", in particular highlighting that resetting the optimizer would also bound the update size. This baseline (resetting the optimizer) is precisely the baseline that we compare to throughout the work, Adam-MR. Unfortunately, we believe this also misses the primary message of our paper: if resetting only $t$ is sufficient to bound the update size, why is it necessary to *also* reset the momentum?

[1] Dohare, Shibhansh, Qingfeng Lan, and A. Rupam Mahmood. "Overcoming policy collapse in deep reinforcement learning." Sixteenth European Workshop on Reinforcement Learning. 2023.

---

### Author Response · Authors · 2024-08-11
**Request for engagement**

Thank you to reviewers **37TD** and **uFqs** for their engagement with our rebuttal!

Now the discussion period is almost over, we ask that reviewers **JkaJ** and **SmBt** please also read and respond to our rebuttal. We have provided an extensive response to each of your reviews and are eager for you to update your scores or provide further clarification.

---

### Decision · Program_Chairs · 2024-09-25

**Decision:**

Accept (poster)

**Comment:**

This paper proposes Adam-Rel, a small modification to Adam, that is motivated by the goal of reducing the impact of non-stationarity on policy optimization due to a sudden increase in update norms.

Reviews for this paper are mixed. The authors had concerns regarding the clarity/quality of initial reviews, but the reviewers clarified their opinions in discussion, and also engaged further in closed discussion about the strengths and weaknesses of this paper. The final discussion focused on two key issues: whether large update norms really are the problem, and whether the experimental evidence sufficient to say that the proposed approach is a good general solution.  I think there are reasonable reasons for doubting this paper on both issues.

The authors demonstrate that at least on the problems they tested, update norms do increase at expected times with Adam, and Adam-Rel alleviates this. But this doesn't necessarily imply that fixing the update norms will always improve optimization. It is entirely possible that since the update norms do not increase without bounds in practice, Adam with suitable hyperparameters can adapt well to them. Similarly, it is possible to come up with scenarios where momentum reset (Adam-MR, a competing approach) might be more beneficial than Adam-Rel --- a reviewer suggests a scenario where the direction of the gradient changes. Could it be the case that Adam-MR is better on some problems (that were not tested) while Adam-Rel is better on other problems? Finally, famously tricky-to-get-right implementation details of PPO make it difficult to trust results when they are on a single benchmark (Crafter).

Clearly, we can't be sure without much more experimentation. Moreover, there is an issue in deep RL research that algorithms from many papers can't be trusted due to insufficient rigor in designing experiments, and they turn out to not be useful in carefully controlled larger scale studies. This is the reason reviewers are being extra cautious.

Nevertheless, after careful consideration, I believe this paper should be accepted as it meets the bar for the results to be interesting enough to warrant further investigation by the broader community. My decision is based on the following:
- The authors have made effort to adopt a cautious tone in the paper (e.g. results "suggest" instead of "show"), instead of overly confident claims.
- The limitations of idealized theoretical analysis are made clear and then examined experimentally, again in the spirit of presenting and analyzing evidence.
- Many limitations of proposed approach are clearly discussed, and I expect that the authors would not object to discussing more.

I believe the only thing missing from this paper is an explicit admission that although the paper has spent considerable resources on experiments, RL environments and algorithms are so varied that it is quite possible that the results don't generalize broadly, and further investigation is warranted. I strongly encourage the authors to add a caveat like this to complete the limitations section, in line with their cautious approach throughout the paper.